# Pd-catalyzed fluoro-carbonylation of aryl, vinyl, and heteroaryl iodides using 2-(difluoromethoxy)-5-nitropyridine

Yumeng Liang [1], Zhengyu Zhao[1] & Norio Shibata [1,2 ✉]

Acyl fluorides have recently gained a lot of attention as robust and versatile synthetic tools in synthetic chemistry. While several synthetic routes to acyl fluorides have been reported, a procedure involving direct insertion of the "fluoro-carbonyl" moiety using a single reagent has not yet been realized. Here we report the preparation of acyl fluorides by palladium-catalyzed fluoro-carbonylation of aryl, vinyl, and heteroaryl iodides using 2-(difluoromethoxy)-5-nitropyridine under CO-free conditions. 2-(difluoromethoxy)-5-nitropyridine is a stable, colorless solid that can be used as an alternative to the toxic gaseous formyl fluoride, which is commonly used under fluoride catalysis conditions. A wide variety of acyl fluorides are efficiently and safely obtained in high yield (up to 99%). A broad range of functional groups is tolerated under the optimized reaction conditions and the method can be applied to the late-stage fluoro-carbonylation of structurally complex $C_{sp2}$-iodides, including bioactive derivatives, such as Fenofibrate, Isoxepac, and Tocopherol. Furthermore, the one-pot transformation of aryl-iodides, including drug-like molecules, into the corresponding amides by successive fluoro-carbonylation/amidation reactions, demonstrates the potential synthetic utility of this strategy.

[1] Department of Nanopharmaceutical Sciences & Department of Life Science and Applied Chemistry, Nagoya Institute of Technology, Gokiso, Showa-ku, Nagoya 466-8555, Japan. [2] Institute of Advanced Fluorine-Containing Materials, Zhejiang Normal University, 688 Yingbin Avenue, Jinhua 321004, China. ✉email: nozshiba@nitech.ac.jp

During the last few decades, fluorinated molecules have found widespread applications in pharmaceuticals, agrochemicals, and functional materials[1–13]. Fluorinated organic compounds have also been in high demand as substrates, reagents, and solvents for general organic chemistry[14–19]. Among the plethora of fluorinated compounds, we have been particularly interested in acyl fluorides (R-COFs), especially aroyl fluorides (Ar-COFs)[20]. Due to the inertness of the C–F bond, the properties and reactivity of R-COFs are very different from those of other acyl halides and their equivalents. R-COFs are robust and multiple synthetic tools have been developed that allow easy and convenient access to a wide range of high-value organic compounds based on acyl coupling reactions that use R-COFs as an "Ar-CO" source[21–26], on decarbonylative coupling reactions that use R-COFs as an "Ar" source[27–33], and on fluorination reactions that use R-COFs as an "F" source[34–37] (Fig. 1a). Although the development of new applications for R-COFs and reactions that involve R-COFs have recently gained attention[21–37], especially in the context of transition-metal catalysis, strategies for the synthesis of R-COFs remain somewhat limited[37–53]. Currently, the synthetic routes to R-COFs are categorized into two groups.

The first group, which involves the fluorination of carboxylic acids or their derivatives, including aldehydes via deoxy-fluorinations, halogen-exchange reactions, or C–H activation reactions, is the central area of the traditional research (type **I**, cleavage **1** in Fig. 1b)[38–49]. The other group includes step-wise fluoro-carbonylation reactions of organic halides using a combination of toxic gaseous carbon monoxide (CO)[50–52] or more stable alternative sources of CO, and fluorinating reagents[53] (type **II**, cleavages **1** and **2** in Fig. 1b). While methods of type **I** and type **II** are usually useful, the development of simpler protocols for the generation of R-COFs remains pertinent, especially if one can avoid the use of toxic and or unstable reagents. However, methods for the direct insertion of the "fluoro-carbonyl" moiety. i.e., "F–C=O" using a single reagent has not yet been realized (type **III**, cleavage **2** in Fig. 1b). Although gaseous formyl fluoride (F–C(=O)H), a potential precursor for a "F–C=O" moiety, has been reported[54–56], formyl fluoride is fundamentally impractical due to its instability, potential toxicity, and the difficulties associated with its handling[54]. In fact, formyl fluoride has not yet been used for fluoro-carbonylation reactions, while formylation reactions with formyl fluoride represent an established area of

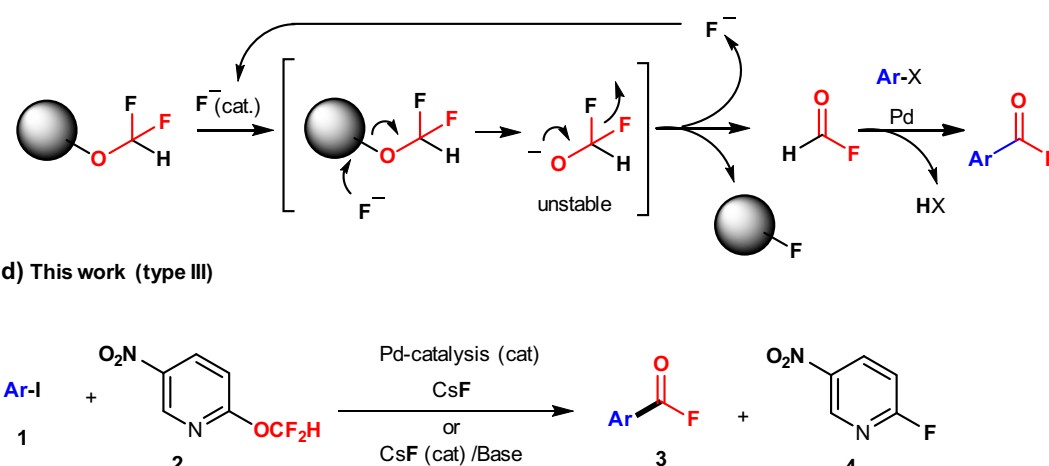

**Fig. 1 Acyl fluorides. a** Synthetic utility, **b** retrosynthesis, **c** conceptual illustration of fluoro-carbonylation reactions, and **d** this work.

**Fig. 2 Fluorocarbonylation with 2.** Synthesis of **2** (**a**) and a photograph of **2** (**b**).

research[54–56]. We thus designed a type **III** strategy that is based on the fluoride-catalyzed in-situ generation of formyl fluoride, followed by a cross-coupling reaction with aryl halides in the presence of a Pd-catalyst. Initially, the difluoromethoxy anion ($-OCF_2H$), should be generated from difluoromethoxy ether under fluoride catalysis, and the resulting difluoromethoxy anion can be expected, given its instability, to spontaneously decompose into formyl fluoride by releasing a fluoride anion ($F^-$), which is responsible for the negative fluorine effect[57,58]. Subsequently, the generated formyl fluoride can be used in cross-coupling reactions with aryl halides under Pd-catalysis (Fig. 1c).

Herein, we report this strategy for the straightforward fluoro-carbonylation of aryl iodides (Ar-I, **1**) by using 2-(difluoromethoxy)-5-nitropyridine (**2**) as both a CO and F source under Pd-catalyzed cross-coupling conditions (Fig. 1d).

The treatment of **1** with **2** in the presence of CsF furnishes the corresponding aroyl fluorides (Ar-COFs, **3**) in good to high yield. The reactions also proceed well using only a catalytic amount of CsF, provided a stoichiometric amount of a base is added. This cross-coupling reaction using **2** works not only for aryl iodides but can also be extended to alkenyl and heteroaryl iodides, which furnishes the corresponding acyl fluorides in good to high yield. R-COFs of pharmaceutical derivatives can also be synthesized under these conditions, despite their often functionalized and complex three-dimensional structures. The key for this fluoro-carbonylation reaction is an in-situ generation of formyl fluoride by decomposition of the unstable $-OCF_2H$, which is delivered from **2** upon a nucleophilic attack of a fluoride-releasing 5-nitropyridine (**4**). Furthermore, we examine the application of this method to the one-pot transformation of aryl iodides into aryl amides, and we investigate the diversification of the resulting aryl fluorides. Moreover, the reaction mechanism is discussed based on the results of control experiments, nuclear magnetic resonance (NMR) spectroscopy, and liquid chromatography–mass spectrometry (LC–MS). As **2** is a stable solid that can be easily synthesized and stored, the method represents a powerful addition to the toolkit of fluoro-carbonylation reactions.

## Results and discussion

**Optimization of the reaction conditions**. 2-(Difluoromethoxy)-5-nitropyridine (**2**) was readily prepared in 83–90% yield from 2-hydroxy-5-nitro-pyridine (**5**) by difluoromethylation[59] in MeCN (rt; 30 min) using the commercially available 2,2-difluoro-2-(fluorosulfonyl)acetic acid (**6**, Chen's reagent[60]) in the presence of NaH (Fig. 2). Compound **2** is an air- and moisture-stable colorless solid and can be treated without special care of handling.

We began our investigation with the reaction between 4-iodobiphenyl (**1a**) and **2** in *N,N*-dimethylformamide (DMF) at 70 °C in the presence of CsF, Pd(OAc)$_2$ (10.0 mol%), and PPh$_3$ (10.0 mol%), which afforded **3a** in 73% yield (Table 1, entry 1). This fluoro-carbonylation was not observed in the absence of Pd (OAc)$_2$, while in the absence of PPh$_3$ the yield was low (entries 2 and 3). The Pd:PPh$_3$ ratio affects the transformation (entries

4–6), and we discovered that a 1:3 ratio affords the best results (entry 5). We then examined different Pd catalysts for this transformation (Supplementary Table 2) and found that Pd (TFA)$_2$ (TFA: trifluoroacetate) is the most effective catalyst, which furnishes **3a** in 95% yield (entry 7). Subsequently, we carried out a screening of phosphine ligands, including monodentate (entries 8–11) and bidentate phosphine ligands (entries 12–15). The yield was improved to 99%, when Xantphos was used (entry 15). The Pd-loading could also be lowered, and the best conditions were determined as **2** (1.2 equiv), CsF (1.5 equiv), Pd(TFA)$_2$ (1.0 mol%), and Xantphos (1.5 mol%) in DMF, which affords **3a** in 99% yield (entry 16). More details for the optimization of the reaction conditions are shown in Supplementary Tables 1–4.

**Substrate scope**. With the optimal reaction conditions in hand, we investigated the substrate scope of the reaction with respect to aryl iodides (Fig. 3). Iodobenzene (**1b**) provided the corresponding product (**3b**) in 92% yield. Both electron-rich and -poor aryl substituents are compatible with the reaction conditions, providing the desired products (**3c–3n**) in generally good to excellent yield. *Meta*-substituted aryl iodides (**1j–1k**) afforded the desired products (**3j–3k**) in high yield. Sterically hindered *ortho*-substituted **1l** provided **3l** in good yield without hampering the reactivity. It should be noted here that the procedure was also efficient for alkenyl iodides (**1o, 1p**), which provided **3o** and **3p** in excellent yield. Heterocyclic aryl iodides (**1q–1t**) can also be used and generate the desired products (**3q–3t**) in good to excellent yield; the results of other heterocyclic aryl substituents are discussed later (vide infra; cf. "Synthetic application"). Reactions of α-iodostyrene (**1u**) and the aliphatic olefin substrate **1v** also proceeded smoothly and afforded the desired products (**3u, 3 v**) in acceptable yield. Due to the hydrolysis of the products during purification and the volatility of some products, the isolated product yields are usually lower than the $^{19}F$ NMR yields, albeit that isolation is possible via column chromatography on silica gel. Interestingly, the reaction can also be scaled up; when the reaction was carried out on a 4.5-mmol scale, **3a** was isolated in 90% yield (Fig. 2b).

To highlight the synthetic utility of this procedure, we used **2** for the late-stage fluoro-carbonylation of natural products and bioactive molecules derivatives. As shown in Fig. 2c, menthol was functionalized to afford **3w** in 35% yield (84% $^{19}F$ NMR yield). Fenofibrate, a synthetic phenoxy-isobutyric acid derivate and prodrug with antihyperlipidemic activity, the fluoro-carbonylation of a fenofibrate derivative **1x** furnished **3x** in 40% yield (88% $^{19}F$ NMR yield). Estrone, arguably one of the most important mammalian estrogens, was transformed into **3y** and **3z** in good yield. Isoxepac, an anti-inflammatory with analgesic and antipyretic activity, afforded **3za** in 63% (87% $^{19}F$ NMR yield). Tocopherol, which exhibits antioxidant activity, could also be fluoro-carbonylated to generate **3zb** in 75% (93% $^{19}F$ NMR yield). The fluoro-carbonylation of a testosterone derivative furnished the desired fluoroacylated product (**3zc**) in 15% (61% $^{19}F$ NMR yield).

**Table 1 Optimization of the reaction conditions for the fluoro-carbonylation of 1 and 2.**

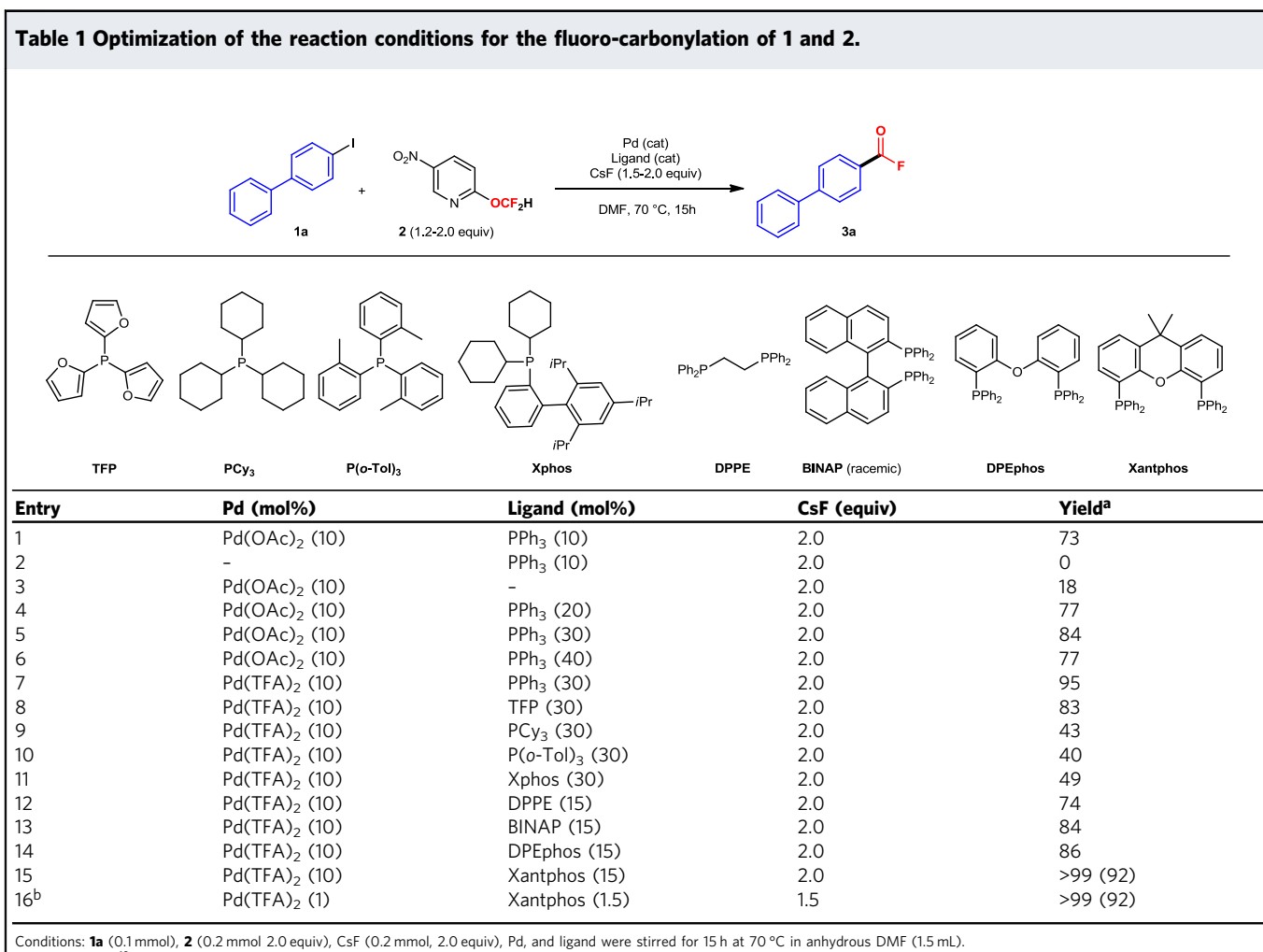

| Entry | Pd (mol%) | Ligand (mol%) | CsF (equiv) | Yield[a] |
|---|---|---|---|---|
| 1 | Pd(OAc)$_2$ (10) | PPh$_3$ (10) | 2.0 | 73 |
| 2 | – | PPh$_3$ (10) | 2.0 | 0 |
| 3 | Pd(OAc)$_2$ (10) | – | 2.0 | 18 |
| 4 | Pd(OAc)$_2$ (10) | PPh$_3$ (20) | 2.0 | 77 |
| 5 | Pd(OAc)$_2$ (10) | PPh$_3$ (30) | 2.0 | 84 |
| 6 | Pd(OAc)$_2$ (40) | PPh$_3$ (40) | 2.0 | 77 |
| 7 | Pd(TFA)$_2$ (10) | PPh$_3$ (30) | 2.0 | 95 |
| 8 | Pd(TFA)$_2$ (10) | TFP (30) | 2.0 | 83 |
| 9 | Pd(TFA)$_2$ (10) | PCy$_3$ (30) | 2.0 | 43 |
| 10 | Pd(TFA)$_2$ (10) | P(o-Tol)$_3$ (30) | 2.0 | 40 |
| 11 | Pd(TFA)$_2$ (10) | Xphos (30) | 2.0 | 49 |
| 12 | Pd(TFA)$_2$ (10) | DPPE (15) | 2.0 | 74 |
| 13 | Pd(TFA)$_2$ (10) | BINAP (15) | 2.0 | 84 |
| 14 | Pd(TFA)$_2$ (10) | DPEphos (15) | 2.0 | 86 |
| 15 | Pd(TFA)$_2$ (10) | Xantphos (15) | 2.0 | >99 (92) |
| 16[b] | Pd(TFA)$_2$ (1) | Xantphos (1.5) | 1.5 | >99 (92) |

Conditions: **1a** (0.1 mmol), **2** (0.2 mmol 2.0 equiv), CsF (0.2 mmol, 2.0 equiv), Pd, and ligand were stirred for 15 h at 70 °C in anhydrous DMF (1.5 mL).
[a]Determined by $^{19}$F NMR spectroscopy. The numbers in parentheses refer to the isolated yield.
[b]**1a** (0.3 mmol), **2** (0.36 mmol, 1.2 equiv), CsF (0.45 mmol, 1.5 equiv), Pd(TFA)$_2$ (1.0 mol%), and Xantphos (1.5 mol%) were stirred for 15 h at 70 °C in anhydrous DMF (2.0 mL).

**Synthetic application I**. As mentioned in "Introduction", acyl fluorides **3** represent a potent platform for a variety of chemical transformations. To demonstrate the broad synthetic utility of **3**, we carried out eight chemical transformations using **3a** (Fig. 4). Specifically, **3a** was successfully transformed into amide **7a** (95%), ester **8a** (85%), and thioester **9a** (76%) by reaction with the heteroatom nucleophiles aniline, phenol, and p-tolyl-thiol, respectively, in the presence of triethylamine in DMF at rt. A Pd-catalyzed cross-coupling reaction of **3a** with PhB(OH)$_2$ using Pd(OAc)$_2$ (2.5 mol%) and PCy$_3$ (10.0 mol%) in the presence of KF in toluene at 120 °C furnished phenyl-coupling product **10a** in 47% yield[22]. A reduction of **3a** with NaBH$_4$ afforded alcohol **11a** in 93% yield, while carboxylic acid **12a** was obtained in 63% from the hydrolysis in water under reflux. The Pd-catalyzed transformation of Ar-COF **3a** with HSiEt$_3$ in toluene at 100 °C in the presence of different phosphine ligands such as PCy$_3$[23] or 1,2-ethanediylbis(dicyclohexylphosphine) (DCPE)[23] resulted in the formation of Ar-CHO **13a** and Ar-H **14a**, respectively, in good to high yield.

**Synthetic application II**. Since the reaction conditions for these fluoro-carbonylation reactions are relatively mild, we examined a one-pot synthesis of amides **7** from aryl iodides **1** via a fluoro-carbonylation/amidation (Fig. 5). For that purpose, para-nitro-phenyl iodide (**1zd**), para-cyano-phenyl iodide (**1ze**), ortho-iodo-pyridine (**1zf**), meta-iodo-pyridine (**1zh**), and para-iodo-pyridine (**1zi**) were

treated individually with **2** under the optimized conditions (Table 1, entry 16). After the completion of the initial fluoro-carbonylation reaction (15 h), fluoro-carbonylation products **3zd**–**3zh** were treated without workup with aniline (PhNH$_2$) and triethylamine (Et$_3$N). After stirring overnight at rt, the desired aryl and heteroaryl amides (**7zd**–**7zh**) were obtained in moderate to good yield. The low yield of **7zh** can be rationalized in terms of the low stability of **1zh**. The aforementioned natural product and bioactive molecule (**1zi**, **1zj**) can also be used in this one-pot fluoro-carbonylation/amidation procedure to furnish the corresponding amides (**7zi**, **7zj**) in good yield.

**Proposed reaction mechanism**. To shed light on the underlying reaction mechanism, we examined a series of experiments under reaction conditions that are slightly different from the optimal conditions (entry 1, Table 2). Initially, we carried out the reaction under the optimized conditions: **2** (1.2 equiv), CsF (1.5 equiv), Pd(TFA)$_2$ (1.0 mol%), and Xantphos (1.5 mol%) in DMF, but using a catalytic amount of CsF (10 mol%). This dramatically decreased the yield of **3a** to 10% (entries 1 and 2), albeit that the yield was recovered to 70% (entry 3) in the presence of a stoichiometric amount of Cs$_2$CO$_3$. Stoichiometric amounts of organic bases such as Et$_3$N or N,N-dimethyl-4-aminopyridine (DMAP) are also effective for this transformation in the presence of a catalytic amount of CsF to furnish **3a** in 51 and 79% yield, respectively (entries 4 and 5). These results suggest that the fluoride in **3a** stems from **2**, not from CsF. Subsequently, we changed the order

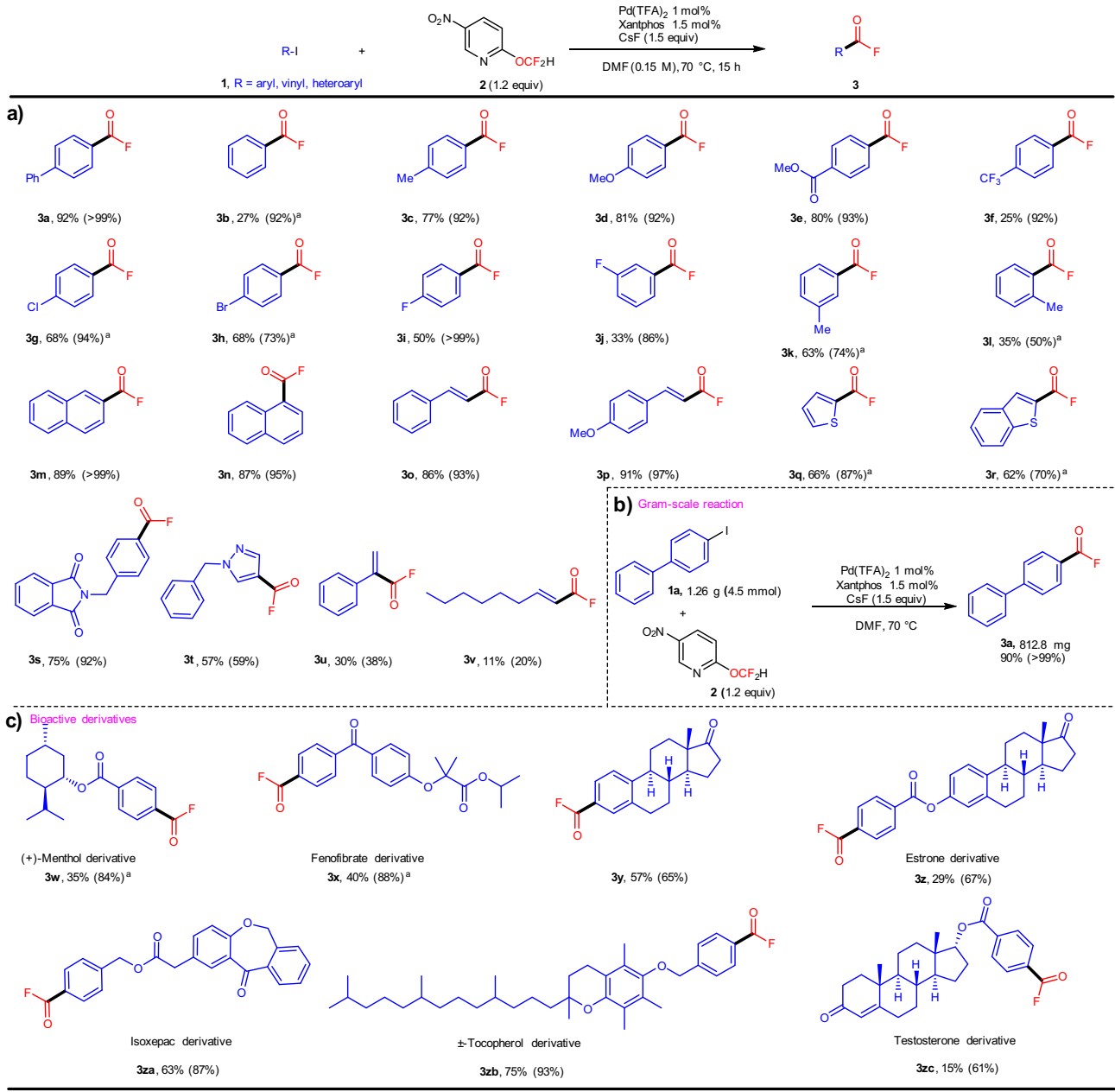

**Fig. 3 Substrate scope and gram-scale reaction of 3a.** Yield values refer to products isolated on a 0.3 mmol scale; yield values in parentheses were determined by $^{19}$F NMR spectroscopy. [a]**2** (0.45 mmol, 1.5 equiv) and CsF (0.6 mmol, 2.0 equiv) was used.

of addition of the reagents (entries 6 and 7). When **1a** was first treated with Pd(TFA)$_2$ (1.0 mol%) and Xantphos (1.5 mol%) at 70 °C for 5 h in DMF, and then with **2** (1.2 equiv) and CsF (1.5 equiv) at 70 °C for another 5 h in DMF, **3a** was obtained in 97% yield (entry 6). However, only 6% of **3a** was detected when the order of addition was reversed, i.e., when **2** was treated with CsF, Pd(TFA)$_2$, and Xantphos in DMF at 70 °C for 5 h, followed by the addition of **1a** (entry 7). Since the optimized reaction conditions (entry 1, Table 2) refer to a reaction where all reagents are mixed from the beginning, it can be concluded that the reaction of **1a** with the Pd-catalyst is much faster than the reaction of formyl fluoride with the Pd-catalyst.

Based on these experiments, additional $^{19}$F NMR experiments, and mass spectroscopy analyses (for details, see Supplementary Figs. 26–28) as well as information from the literature[60], we would like to propose a plausible reaction mechanism (Fig. 6). Reaction mechanism starts with the

generation of a phosphine-ligated Pd(0) species (LnPd$^0$), which undergoes an oxidative addition into the C–I bond of Ar-I (**1a**), resulting in the formation of aryl Pd(II) species **I**. An LC–MS analysis supported the generation of **I** by confirming the presence of Pd-Xantphos species **I′** ($m/z = 837$) and **I″** ($m/z = 731$). The process from LnPd$^0$ to **I** under concomitant detection of **I′** and **I″** is in good agreement with the report by Lee and Morandi[61]. The resulting complex **I** can then coordinate to the formyl fluoride, generated from **2** via a fluoride-catalyzed self-decomposition of the difluoromethoxy anion, to furnish I–Pd–Ar species **II**. Then, the insertion of the aryl group across the C=O moiety in Pd-complex **II** providing intermediate **III**, followed by a base-induced β-hydride elimination would directly afford **3a** under regeneration of the Pd(0) catalyst. Related pathways, involving β-hydride elimination steps for cross-coupling reactions, have been reported by Martin (Pd-catalysis)[62], Newman (Ni-catalysis)[63], and Lee

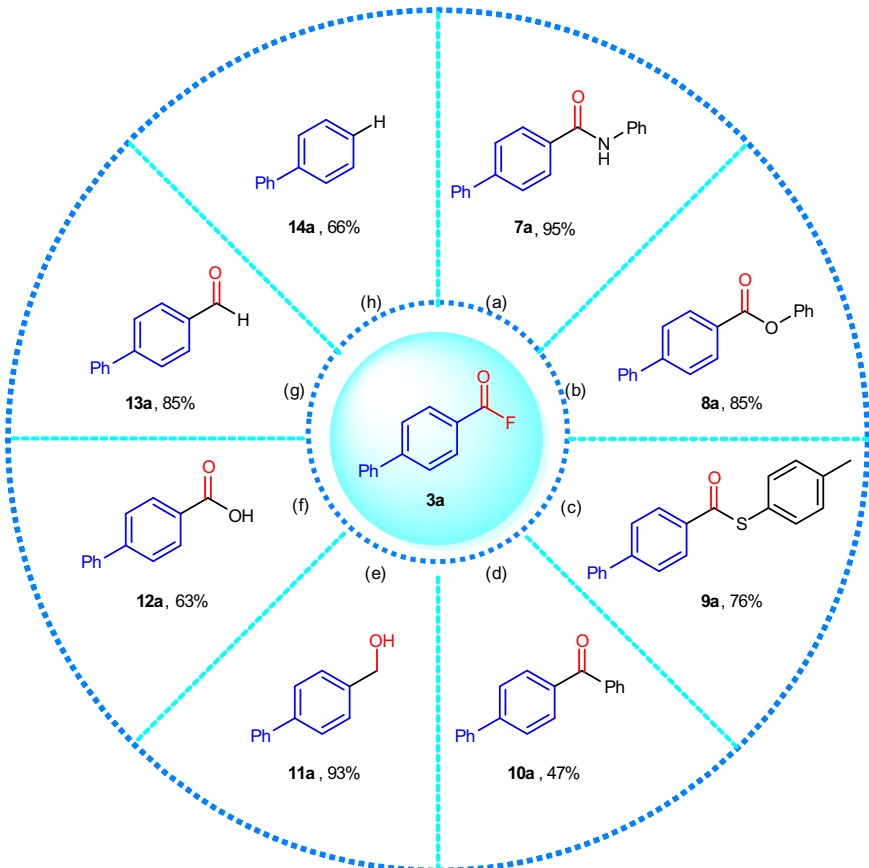

**Fig. 4 Chemical diversification of 3a.** Reaction conditions: **a** PhNH₂ (2.0 equiv), NEt₃ (3.0 equiv), DMF, rt. **b** PhOH (1.2 equiv), NEt₃ (2.0 equiv), DMF, rt. **c** 4-Me-PhSH (1.2 equiv), NEt₃ (2.0 equiv), DMF, rt. **d** PhB(OH)₂ (1.5 equiv), Pd(OAc)₂ (2.5 mol%), PCy₃ (10.0 mol%), KF (1.5 equiv), toluene, 120 °C. **e** NaBH₄ (1.0 equiv), iPrOH, rt. **f** H₂O, reflux. **g** HSiEt₃ (1.4 equiv), Pd(OAc)₂ (2.5 mol%), PCy₃ (7.5 mol%), toluene, 100 °C. **h** HSiEt₃ (1.4 equiv), Pd(OAc)₂ (2.5 mol%), DCPE (3.8 mol%), toluene, 100 °C. For full experimental details, see Supplementary Figs. 10–17.

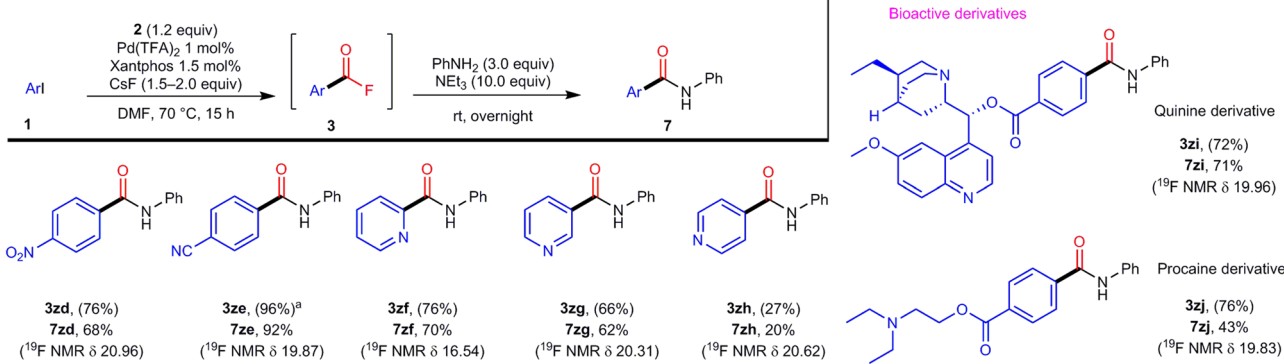

**Fig. 5 One-pot amidations of 1 to afford 7.** Yield values refer to products **7** isolated on a 0.3 mmol scale. Yield values in parentheses refer to the yield of intermediates **3** as determined by a ¹⁹F NMR spectroscopic analysis of the reaction mixture without a work-up procedure. ᵃ**2** (0.45 mmol, 1.5 equiv) and CsF (0.6 mmol, 2.0 equiv) was used. For full experimental details, see Supplementary Figs. 18–24.

(Ni-catalysis)[64]. However, the details of the reaction mechanism remain to be determined.

In summary, we have developed an efficient strategy for the Pd-catalyzed fluoro-carbonylation of aryl, vinyl, and heteroaryl iodides using formyl fluoride that is generated spontaneously from 2-(difluoromethoxy)-5-nitropyridine (**2**). The high reactivity and broad applicability of this synthetic methodology suggest that this protocol may become a compelling alternative synthetic route to acyl fluorides, which represent essential intermediates in the process of pharmaceutical integration. So far, four methods for the Pd-catalyzed (or mediated) fluoro-carbonylation have been reported using toxic CO (Tanaka[50], Kiji[51], Hiyama[52]) or a stable CO-equivalent (Manabe[53]) with different combinations of fluoride sources; in comparison, our method exhibits a substantially broader substrate scope and uses **2** as a combined source of CO and fluoride. Further investigations into the extension of this fluoro-carbonylation strategy to generate more complex substrates, as well as establishing the details of the reaction mechanism, are currently in progress in our laboratory.

**Table 2 Mechanistic understanding through catalyst- and base-loading studies and the order of addition of 1a and 2.**

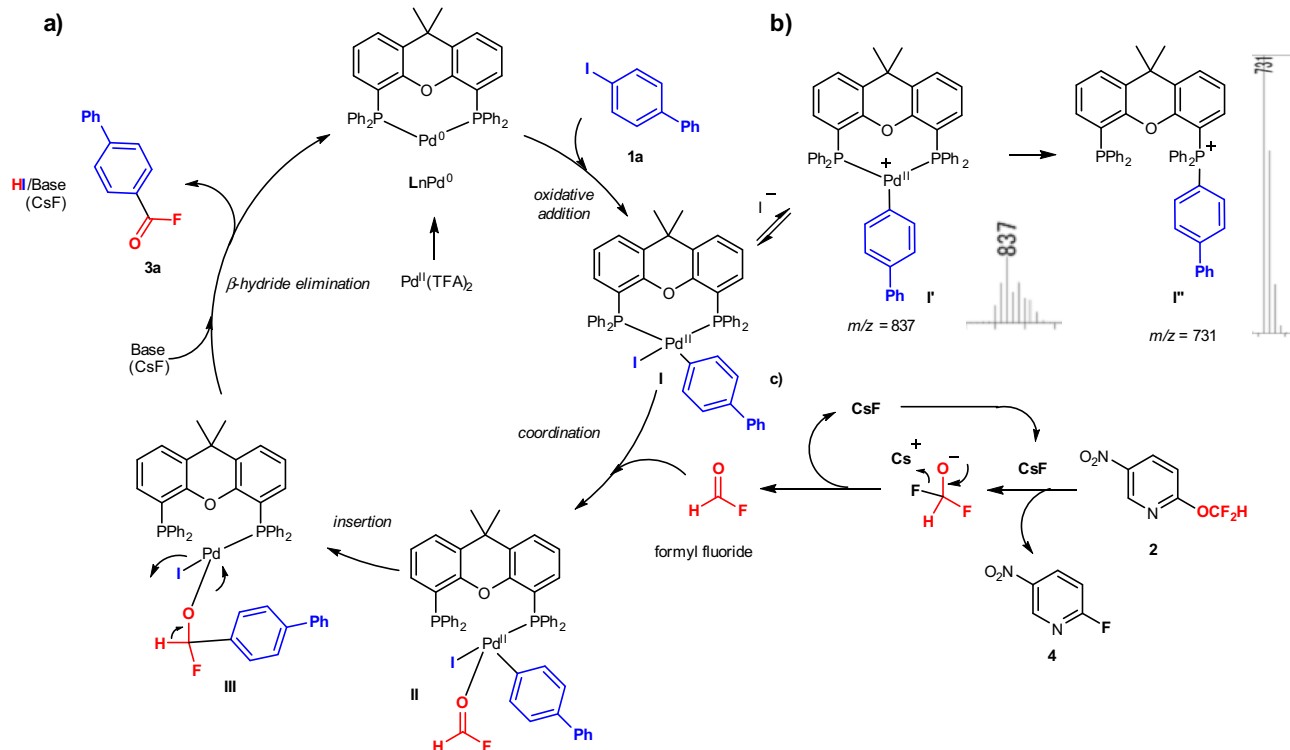

| Entry[a] | CsF (equiv) | Base (equiv) | Yield (%)[b] |
|---|---|---|---|
| 1 | 1.5 | – | >99 |
| 2 | 0.1 | – | 10 |
| 3 | 0.1 | Cs$_2$CO$_3$ (1.0) | 70 |
| 4 | 0.1 | Et$_3$N (2.0) | 51 |
| 5 | 0.1 | DMAP (2.0) | 79 |
| 6[c] | 1.5 | – | >97 |
| 7[d] | 1.5 | – | 6 |

[a]**1a** (0.3 mmol), **2** (0.36 mmol, 1.2 equiv), CsF, base, Pd(TFA)$_2$ (1.0 mol%), and Xantphos (1.5 mol%) were stirred for 15 h at 70 °C in anhydrous DMF (2.0 mL).
[b]$^{19}$F NMR yield.
[c]**1a** (0.3 mmol) was stirred in the presence of Pd(TFA)$_2$ (1.0 mol%) and Xantphos (1.5 mol%) at 70 °C. After 5 h of stirring, **2** (0.36 mmol, 1.2 equiv) and CsF (1.5 equiv) were added to the reaction mixture, before stirring was continued for another 5 h.
[d]**2** (0.36 mmol, 1.2 equiv) was stirred at 70 °C in the presence of CsF (1.5 equiv), Pd(TFA)$_2$ (1.0 mol%), and Xantphos (1.5 mol%). After 5 h of stirring, **1a** (0.3 mmol) was added to the reaction mixture, before stirring was continued for another 5 h.

**Fig. 6 A plausible reaction mechanism. a** A proposed catalytic cycle for Pd-catalyzed acyl fluoride synthesis. **b** Key reaction intermediates detected by mass spectrometry (MS). **c** A process for the generation of formyl fluoride catalyzed by CsF.

## Methods

**General procedure for the generation of acyl fluorides 3a using a stoichiometric amount of CsF.** An oven-dried vessel containing a magnetic stirrer bar was charged with Pd(TFA)$_2$ (1.0 mg, 0.003 mmol, 1.0 mol%), Xantphos (2.6 mg, 0.0045 mmol, 1.5 mol%), CsF (68.4 mg, 0.45 mmol, 1.5 equiv), and anhydrous N,N-dimethylformamide (DMF, 2.0 mL, 0.15 M) in a nitrogen-filled glovebox. After stirring the reaction mixture for 10 min at room temperature, **2** (0.36 mmol, 1.2 equiv) and aryl iodide **1a** (0.3 mmol, 1.0 equiv) were added. The vessel was capped with a rubber septum, removed from the glovebox, and stirred for 15 h at 70 °C. Then, the mixture was cooled to room temperature and the yield (>99%) was determined by $^{19}$F NMR analysis of the crude reaction mixture using C$_6$H$_5$F (28.5 μL, 0.3 mmol, 1.0 equiv) as an internal standard. The crude mixture was

directly purified by flash chromatography on silica gel (thickness: 10 cm; diameter: 2 cm) to afford **3a** (55.3 mg, 92% yield) as a white solid.

**General procedure for the generation of acyl fluorides 3a Using a catalytic amount of CsF.** An oven-dried vessel containing a magnetic stirrer bar was charged with Pd(TFA)$_2$ (1.0 mg, 0.003 mmol, 1.0 mol%), Xantphos (2.6 mg, 0.0045 mmol, 1.5 mol%), CsF (4.6 mg, 0.03 mmol, 10.0 mol%), Cs$_2$CO$_3$ (97.7 mg, 0.3 mmol, 1.0 equiv), and anhydrous DMF (2.0 mL, 0.15 M) in a nitrogen-filled glovebox. After stirring the reaction mixture for 10 min at room temperature, **2** (0.36 mmol, 1.2 equiv) and aryl iodide **1a** (0.3 mmol, 1.0 equiv) were added. The vessel was capped with a rubber septum, removed from the glovebox, and stirred

for 15 h at 70 °C. Then, the mixture was cooled to room temperature, and the yield (70%) was determined by $^{19}$F NMR analysis of the crude reaction mixture using $C_6H_5F$ (28.5 μL, 0.3 mmol, 1.0 equiv) as an internal standard.

**General procedure for the one-pot transformation of 1 into amides 7zd**. An oven-dried vessel containing a magnetic stirrer bar was charged with Pd(TFA)$_2$ (1.0 mg, 0.003 mmol, 1.0 mol%), Xantphos (2.6 mg, 0.0045 mmol, 1.5 mol%), CsF (68.4 mg, 0.45 mmol, 1.5 equiv), and anhydrous DMF (2.0 mL, 0.15 M) in a nitrogen-filled glovebox. After stirring the reaction mixture for 10 min at room temperature, **2** (0.36 mmol, 1.2 equiv) and aryl iodide **1zd** (0.3 mmol, 1.0 equiv) were added. The vessel was capped with a rubber septum, removed from the glovebox, and stirred for 15 h at 70 °C. Then, the mixture was cooled to room temperature, before NEt$_3$ (418 μL, 3.0 mmol, 10.0 equiv) and PhNH$_2$ (81 μL, 0.9 mmol, 3.0 equiv) were added and stirring was continued overnight at room temperature. After quenching with H$_2$O (20 mL), the mixture was extracted with AcOEt (3 × 20 mL) and the combined organic layers were dried over anhydrous Na$_2$SO$_4$. After filtration, the filtrate was concentrated under reduced pressure. The crude residue was purified by flash chromatography on silica gel (eluent: *n*-Hexane: AcOEt = 1:1, v/v) to afford **7zd** (45.8 mg, 63% yield) as a pale yellow solid.

The NMR yield of \*3zd (76%) was directly determined by $^{19}$F NMR analysis of the crude reaction mixture using $C_6H_5F$ (28.5 μL, 0.3 mmol, 1.0 equiv) as an internal standard.

## Data availability

The data supporting the findings of this study are available within the paper and its Supplementary Information. All relevant data are also available from the authors.

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

## Acknowledgements
This work was supported by JSPS KAKENHI Grants JP 18H02553 (KIBAN B) and JP 18H04401 (Middle Molecular Strategy). The authors also would like to thank Tosoh Finechem Corporation for their support.

## Author contributions
N.S. conceived the concept of this study. Y.L. optimized the reaction conditions and surveyed the substrate scope. Y.L. and Z.Z. prepared the starting materials. N.S. directed the project. N.S. and Y.L. prepared the manuscript.

## Competing interests
The authors declare no competing interests.
