## [Peer Review File · Communications Chemistry]

REVIEWERS' COMMENTS:

Reviewer #1 (Remarks to the Author):

acyl fluorides are important synthetic intermediates. The current procedure using 2-(difluoromethoxy)-5-nitropyridine as the source of COF. Compared with known procedures, such as CO source and CsF, the current procedure has no advantage. Additionally, the atom efficiency is very low. Moreover, the title and content is misleading. The content has nothing to do with carbonylation, as no CO was generated or involved. Hence, I can not support the acceptance of this manuscript in the current form.

Reviewer #2 (Remarks to the Author):

The authors describe the Pd-catalyzed fluoro-carbonylation using a difluoromethoxypyridine derivative that they have developed as an easy-to-handle surrogate of formyl fluoride. A wide variety of acyl fluorides were successfully synthesized with good functional-group compatibility. Successful transformations of the products into other derivatives indicate the usefulness of the reaction. I think that this work deserves publication, although some minor revisions are required as shown below.

(1) Scheme 3, compounds 3y and 3z: One of the hashed wedged bonds (C-C bond in ring B) looks strange.

(2) Page 4, the 2nd paragraph: The statement "Fenofibrate furnished 3x in 40% yield" is misleading, because fenofibrate itself has a chloro group instead of iodo group at the reactive position. The authors used the iodo version of fenofibrate instead of fenofibrate itself. So, the statement should be corrected.

(3) Table 2, footnote a: "2" of "Pd(TFA)₂" must be a subscript.

Reviewer #3 (Remarks to the Author):

This paper reports the palladium-catalyzed introduction of a fluoro carbonyl group into aroyl/vinyl iodides with 2-(difluoromethoxy)-5-nitropyridine. The authors clarified substrate scope and its application of formed aroyl fluorides to various derivatives and proposed in-situ formation of formyl fluoride in a plausible mechanism. This manuscript deserved to be published in this journal.

Concerning the following explanation, quite recently, the group has reported a novel finding.

Page 1, lines 41-42

「and on fluorination...R-COFs as an "F" source³⁵⁻³⁷ (Scheme 1a).」

Page 1, lines 44-45

「strategies for the synthesis of R-COFs remain somewhat limited.³⁸⁻⁵³」

Therefore, the reference shown below should be cited.

Organometallics 2020, 39, 856-861.

Reviewer #1 (Remarks to the Author):

acyl fluorides are important synthetic intermediates. The current procedure using 2-(difluoromethoxy)-5-nitropyridine as the source of COF. Compared with known procedures, such as CO source and CsF, the current procedure has no advantage. Additionally, the atom efficiency is very low. Moreover, the title and content is misleading. The content has nothing to do with carbonylation, as no CO was generated or involved. Hence, I can not support the acceptance of this manuscript in the current form.

Answer: This could be happened by the misunderstanding. First, the introduction part thoroughly gives the current state of the art, including the significant challenges, of existing fluoro-carbonylation methods for comparisons. Our catalytic method uses low catalyst loading of Pd/XantPhos, operates at mild reaction conditions, and without the use of carbon monoxide, which is a significant advantage. Second, a diverse set of aryl iodides were converted to acid fluorides in good to high yields. Third (most important), the fundamental aspect of our work is the utility of formyl fluoride as the “fluoro-carbonylating reagent” that is in-situ generated from the base-promoted decomposition of 2-(difluoromethoxy)-5-nitropyridine. This strategy is a unique approach towards the synthesis of acid fluorides and other carbonyl compounds (such as amides, as shown in a telescopic reaction).

I hope the Reviewer #1 could kindly read the manuscript carefully, since we clearly wrote the importance and novelty of the manuscript.

Reviewer #2 (Remarks to the Author):

The authors describe the Pd-catalyzed fluoro-carbonylation using a difluoromethoxypyridine derivative that they have developed as an easy-to-handle surrogate of formyl fluoride. A wide variety of acyl fluorides were successfully synthesized with good functional-group compatibility. Successful transformations of the products into other derivatives indicate the usefulness of the reaction. I think that this work deserves publication, although some minor revisions are required as shown below.

Answer: Thank you very much for your high evaluation. I revised the manuscript according to your suggestions.

Comment 1: Scheme 3, compounds 3y and 3z: One of the hashed wedged bonds (C-C bond in ring B) looks strange.

Answer: Revised.

Comment 2: The statement “Fenofibrate furnished 3x in 40% yield” is changed to “Fenofibrate, a synthetic phenoxy-isobutyric acid derivate and prodrug with antihyperlipidemic activity, the fluoro-carbonylation of a fenofibrate derivative 1x furnished 3x in 40% yield (88% ¹⁹F NMR yield).”

Answer: Revised.

Reviewer #3:

This paper reports the palladium-catalyzed introduction of a fluoro carbonyl group into aroyl/vinyl iodides with 2-(difluoromethoxy)-5-nitropyridine. The authors clarified substrate scope and its application of formed aroyl fluorides to various derivatives and proposed in-situ formation of formyl fluoride in a plausible mechanism. This manuscript deserved to be published in this journal.

Answer: Thank you very much for your high evaluation. I revised the manuscript according to your suggestions.

Concerning the following explanation, quite recently, the group has reported a novel finding.

Page 1, lines 41-42

「and on fluorination...R-COFs as an “F” source³⁵⁻³⁷ (Scheme 1a).」

Page 1, lines 44-45

「strategies for the synthesis of R-COFs remain somewhat limited.³⁸⁻⁵³」

Therefore, the reference shown below should be cited.

Organometallics 2020, 39, 856-861.

Answer: Revised.